# Characterization of a Novel Superoxide Dismutase from a Deep-sea Sea Cucumber (*Psychoropotes verruciaudatus*)

**DOI:** 10.3390/antiox12061227

**Published:** 2023-06-07

**Authors:** Yanan Li, Zongfu Chen, Peng Zhang, Feng Gao, Junfeng Wang, Li Lin, Haibin Zhang

**Affiliations:** 1Guangzhou Key Laboratory of Aquatic Animal Diseases and Waterfowl Breeding, College of Animal Sciences and Technology, Zhongkai University of Agriculture and Engineering, Guangzhou 510222, China; liyanan@zhku.edu.cn (Y.L.); vvzongfuchen@zhku.edu.cn (Z.C.); 3137322717@zhku.edu.cn (F.G.); linli@zhku.edu.cn (L.L.); 2Guangdong Provincial Key Laboratory of Fishery Ecology and Environment, South China Sea Fisheries Research Institute, Chinese Academy of Fishery Sciences, Guangzhou 510300, China; zhangpeng@scsfri.ac.cn; 3Guangdong Key Laboratory of Marine Materia Medica, South China Sea Institute of Oceanology, Chinese Academy of Sciences, Guangzhou 510301, China; wangjunfeng@scsio.ac.cn; 4Institute of Deep-sea Science and Engineering, Chinese Academy of Sciences, Sanya 572000, China

**Keywords:** Holothuroidea, deep sea, enzyme expression, purification

## Abstract

At present, deep-sea enzymes are a research hotspot. In this study, a novel copper–zinc superoxide dismutase (CuZnSOD) was successfully cloned and characterized from a new species of sea cucumber *Psychropotes verruciaudatus* (PVCuZnSOD). The relative molecular weight of the PVCuZnSOD monomer is 15 kDa. The optimum temperature of PVCuZnSOD is 20 °C, and it maintains high activity in the range of 0–60 °C. It also has high thermal stability when incubated at 37 °C. PVCuZnSOD has a maximum activity of more than 50% in the pH range of 4–11 and a high activity at pH 11. In addition, PVCuZnSOD has strong tolerance to Ni^2+^, Mg^2+^, Ba^2+^, and Ca^2+^, and it can withstand chemical reagents, such as Tween20, TritonX-100, ethanol, glycerol, isopropanol, DMSO, urea, and GuHCl. PVCuZnSOD also shows great stability to gastrointestinal fluid compared with bovine SOD. These characteristics show that PVCuZnSOD has great application potential in medicine, food, and other products.

## 1. Introduction

Reactive oxygen species (ROS), including peroxides, superoxides, hydroxyl radical, singlet oxygen, and alpha-oxygen, are natural by-products of the normal metabolism of biological oxygen and play an important role in cell signaling and homeostasis [1]. However, when the body is in an unfavorable environment, ROS in the body may increase dramatically, causing serious damage to the body, which is known as oxidative stress [2]. Oxidative stress is closely related to various pathological processes in the body [3], and causes severe damage to cell components, DNA damage or mutations, and cell death [4]. In defending against oxidative stress, aerobic organisms have developed a well-developed oxidative defense system. Antioxidative enzymes, including superoxide dismutase (SOD), thioredoxin peroxidase, glutathione peroxidase, and catalase [5], play a crucial role in this system, among which SODs are the most important.

Based on the effects of SODs on antioxidant stress, they have extraordinary therapeutic functions. For example, chronic wounds of diabetes patients often cause serious oxidative stress, which seriously hinders wound healing. Dressing with added SODs can effectively promote the repair of chronic wounds of diabetics by promoting wound closure and collagen deposition [6]. Obesity-related metabolic syndrome is closely related to insulin resistance caused by oxidative stress, and adding melon SOD to the daily diet can help reduce weight and insulin resistance [7]. SODs can also treat arthritis in rats and reduce liver injury in a rat liver ischemia/reperfusion model [8]. In addition, SODs are also widely used in skin management. Oxidative stress can accelerate skin aging. Skincare products containing SODs can help alleviate oxidative stress and promote the improvement of skin structure and function, and thus play an anti-aging and anti-wrinkle role [9]. In summary, SODs are widely used in medicine, food, and cosmetics [10]. At present, a large number of SOD products have been developed, such as manganese sulfide nanoparticles embedded in superoxide dismutase that can be used to alleviate Parkinson’s disease [11], health drinks containing SOD [12], and skin nutrients with the anti-aging effect of SOD [9].

However, poor dynamic stability and easily degradable SODs have limited their further application. To solve those issues, researchers have attempted to modify or mutate SODs, or search for SODs with excellent performance. SODs with superior properties have been successfully characterized from various organisms, such as the SODs of *Caiman latirostris* [13], *Zostera marina* [14], and *Acipenser baerii* [15]. In recent years, deep-sea biomes have attracted the attention of researchers. Li et al. [16] successfully characterized an SOD with kinetically stable, strong enzyme–substrate affinity, and high catalytic efficiency from a hadal sea cucumber in the Mariana Trench. In addition, Aik-Hong Teh et al. [17] characterized an SOD from *Cryptococcus liquefaciens* strain N6 with high tolerance to heavy metals. The SOD from *Geobacillus* sp. EPT3 had excellence thermal stability [18]. The SOD from *Bacillus* sp. SCSIO 15121 was not sensitive to multiple chemicals [19]. Therefore, SODs from deep-sea biomes may have several excellent properties or special abilities, which may be associated with their environmental adaptation.

*Psychropotes verruciaudatus* is a new species of sea cucumber, which is found in the deep sea at a depth of 2510 m [20]. In this study, CuZnSOD from *P. verruciaudatus* was cloned and expressed, and its stability for potential applications in medicine, food, and the nutraceutical field was preliminarily evaluated.

## 2. Materials and Methods

### 2.1. Materials, Total RNA Extraction, and cDNA Cloning

*P. verruciaudatus* were collected in the South China Sea (112.3685° E 17.9238° N) at a depth of 2510 m. RNeasy Plus universal kits (Qiagen, Hilden, Germany) were used to extract total RNA following the manufacturer’s instructions. Then, total RNA was sequenced by Novogene (Tianjin, China). Following the manufacturer’s instructions, first-strand cDNA was synthesized by using the Takara reverse transcription kit (PrimeScript™ II 1st strand cDNA synthesis kit, Takara, Japan).

### 2.2. Sequence Analysis of PVCuZnSOD

The similarity of nucleotide sequence was searched by using the BLAST algorithm (http://www.ncbi.nlm.nih.gov/blast, accessed on 15 February 2023). Then, the ORF finder (http://bioinf.ibun.unal.edu.co/servicios/sms/orf_find.html, accessed on 15 February 2023) was used to determine the open reading frame of PVCuZnSOD, and the SMART program (https://smart.embl.de/smart/set_mode.cgi?NORMAL=1, accessed on 15 February 2023) was used to identify the domain structure of PVCuZnSOD. The signal peptide was predicted by SignalP4.1 (https://services.healthtech.dtu.dk/service.php?SignalP-4.1, accessed on 15 February 2023). Jalview and Expasy PROSTIE (https://www.expasy.org/resources/prosite, accessed on 15 February 2023) were used to predetermine the second structure and to scan the motif, respectively. Based on the most appropriate model (PDB ID: 7wx0.1.A Superoxide dismutase [Cu-Zn], E40K variant of Cu/Zn-superoxide dismutase from dog), the three-dimensional model was built by SWISSMODEL (http://swissmodel.expasy.org, accessed on 15 February 2023), and PyMOL 2.5.3 was used to visualize the protein structure. Sequence alignment and phylogenetic tree were performed by MEGA11 [21]. The reliability of the branch was tested with 1000 repetitions.

### 2.3. Recombinant Protein Expression

Based on the manufacturer’s instruction, PrimeSTAR^®^ GXL DNA Polymerase (Takara, Japan) was used to amplify PVCuZnSOD by polymerase chain reaction (PCR) with the primers F: CGGGATCCATGTCTGTCGACGCAGTA and R: GCTGCAGTTATGCTTTCTTAATGCC. Subsequently, the PCR product and pColdII contained His-Tag (Takara, Japan) were digested with *Bam*H1 and *Pst*I (Takara, Japan). Then, T4 DNA Ligase (Takara, Japan) was used to construct the expression vector of pColdII-PVCuZnSOD. After the reaction, the reaction tube was transferred to an ice-water bath for 5 min. pColdII-PVCuZnSOD was transformed into competent cells (*Escherichia coli* DH5α, Takara, Japan). The cell containing the recombinant plasmid was selected at 37 °C using a LB plate added with ampicillin (100 μg/mL).

### 2.4. Extraction and Purification of Recombinant Protein

Cells containing pColdII-PVCuZnSOD vector were cultivated at 37 °C, 200 rpm, in LB broth (Amp^+^) until the cell density reached OD600 of 0.4–0.6. Then 1 mM IPTG (final concentration) was added and continuously cultivated at 16 °C for 24 h. Cells were harvested by centrifugation at 4200× *g* and 4 °C for 35 min. The supernatant was discarded, and the pellet was washed two times with PBS, and centrifuged at 4200× *g* and 4 °C for 10 min. Then, the pellet was resuspended with 10 mL of PBS and added with PMSF (1:100). Subsequently, the mixture was sonicated for 1 h (sonication operated continuously for 5 s with an interval of 3 s.) in an ice bath and transferred to a tube for centrifugation at 15,000× *g* and 4 °C for 20 min.

The supernatant was transferred to Ni-NTA agarose column, which was pre-equilibrated by PBS, and this process was repeated five times. The column was re-washed with PBS and imidazole at a final concentration of 20, 40, and 60 mM, in turn. Then, the bound proteins were eluted with elution buffer (50 mM Tris-HCL, 50 mM NaCl, 0.1 mM EDTA, and 300 mM imidazole, pH 8.0). Imidazole was removed by 24 h dialysis. Protein concentration was calculated by using the improved BCA protein concentration assay kit following the manufacturer’s instructions. The purification and expression effects of recombinant PVCuZnSOD was evaluated using 12% SDS-PAGE.

### 2.5. Western Blot Analysis

The recombinant protein was mixed with loading buffer, boiled at 100 °C for 5 min, centrifuged at 12,000× *g* for 1 min, and separated by 12% SDS-PAGE gel before being transferred to a PVDF membrane (Burlington, MA, USA) by a trans-blot SD semi-dry electrophoretic transfer cell (Beijing Liuyi, China). Then, the PVDF membrane was incubated with 5% BSA for 2 h, primary antibody (Anti-His Tag Mouse Monoclonal Antibody, Abcam, Cambridge, UK) overnight, and secondary antibody (HRP Goat Anti-Mouse lgG, Abcam, Cambridge, UK) for 2 h. Each time the incubation solution was changed, 1× TBST was used to wash the PVDF membrane. Finally, the target band on the PVDF membrane was dyed by using the Pierce™ ECL Plus Western blotting substrate (ThermoFisher, Waltham, MA, USA) and detected by using the chemiluminescent imaging system.

### 2.6. PVCuZnSOD Assay

The activity of PVCuZnSOD was assessed by the total superoxide dismutase assay kit (Nanjing Jiancheng Bioengineering Institute) following the manufacturer’s instructions, which was used in all assessments except for the determination of kinetic parameters. Kinetic parameters were determined using the riboflavin–NBT method. In brief, the enzymatic activity was evaluated by detecting the inhibition of PVCuZnSOD on the reduction of nitro-blue tetrazolium under light. The 3 mL reaction system includes 2.05 mL of PBS, 0.3 mL of 220 mM methionine solution, 0.3 mL of 1.25 mM NBT solution, 0.3 mL of riboflavin solution with different concentrations, and 0.05 mL of protein sample. The reaction system was incubated under light for 30 min, and then absorbance at 560 nm was detected. The control groups used PBS instead of protein samples. Three replicates were prepared for each measurement.

### 2.7. Effect of Temperature on the Activity of PVCuZnSOD

A hydroxylamine method was used in the assay. The samples were incubated at different temperatures (4 °C, 10 °C, 20 °C, 30 °C, 40 °C, 50 °C, 60 °C, and 70 °C) for 15 min. Then, the residual activity of PVCuZnSOD was assessed (Chapter 2.6). The highest value of enzymatic activity was set as 100%. Other values were divided by the highest value to calculate the remaining activity.

In examining the stability of PVCuZnSOD at different temperatures, the protein solution was incubated at different temperatures (4 °C, 20 °C, 37 °C, and 50 °C), and enzymatic activity was checked at 0, 3, 6, 9, 12, 24, and 36 h. The enzymatic activity of PVCuZnSOD at 0 h was set as 100%.

### 2.8. Effect of pH on the Activity of PVCuZnSOD

The samples were incubated in different buffers (Gly+HCL buffer [pH = 1.0, pH = 2.0, pH = 3.0], NaAc-Hac buffer [pH = 4.0, pH = 5.0], NaH_2_PO_4_-citric acid buffer [pH = 6.0, pH = 7.0], Tris-HCl buffer [pH = 8.0, pH = 9.0], and glycine-NaOH buffer [pH = 10.0, pH = 11.0, pH = 12.0]) for 1 h at 25 °C. Afterward, the residual activity of PVCuZnSOD was assessed using a Nanjing Jiancheng Bioengineering Institute kit. The highest value of enzymatic activity was set as 100%. Other values were divided by the highest value to calculate the remaining activity.

### 2.9. Effect of Metal Ions on the Activity of PVCuZnSOD

The samples were incubated in a buffer containing different final concentrations (0.1, 1, 5, and 10 mM) of metal ion (Mn^2+^, Cu^2+^, Zn^2+^, Co^2+^, Ni^2+^, Mg^2+^, Ba^2+^, and Ca^2+^) chlorides at 25 °C for 30 min. The buffer of the control group did not contain the abovementioned metal ions. The activity of the control group was set as 100%.

### 2.10. Effect of Surfactants on the Activity of PVCuZnSOD

The effect of surfactants (Tween 20, TritonX-100, and SDS) with different concentrations on the activity of PVCuZnSOD was investigated. The samples were incubated with the abovementioned surfactant for 30 min at 25 °C. In addition, the control group used buffer instead of surfactant. The activity of the control group was set as 100%.

### 2.11. Effects of Organic Solvents on the Activity of PVCuZnSOD

Enzymatic solutions containing different final concentrations (5% and 10% [*v*/*v*]) of methanol, ethanol, glycerol, isopropanol, acetone, and DMSO were incubated at 25 °C for 30 min. The control group used buffer instead of organic solvents. The activity of the control group was set as 100%.

### 2.12. Effect of Denaturant on the Activity of PVCuZnSOD

In detecting the stability of PVCuZnSOD in urea and guanidine hydrochloride (GuHCl), enzymatic solutions containing different concentrations (0 M, 1 M, 2 M, 3 M, and 4 M) of urea or GuHCl were incubated at 25 °C for 30 min. The activity of the control groups was set as 100%.

### 2.13. Effect of Digestive Enzyme on the Activity of PVCuZnSOD

In detecting the stability of PVCuZnSOD in intestinal and gastric juice, artificial gastric juice was prepared in accordance with the United States Pharmacopoeia; 2 g of NaCl and 3.2 g of pepsin (1000 U/mg) were added to 1000 mL of water containing 1 M HCl, and the pH was adjusted to 3.0. The samples were incubated with artificial gastric juice for 3 h, and the residual activity was determined at 0, 1, 2, and 3 h. In addition, the sample was incubated with 100 times the mass of the digestive enzyme (trypsin: chymotrypsin = 2400:400) at 37 °C and pH 7.4 for 3 h, and the activity was measured every hour. The activity was set to 100% at 0 h. Then, a PeptideCutter (https://web.expasy.org/peptide_cutter/, accessed on 12 March 2023) was used to predict the cutting sites of digestive enzymes. In addition, bovine SOD (9054-89-1, Sangon Biotech, Guangzhou) was used for the digestive enzyme experiment, and the result was compared with PVCuZnSOD.

### 2.14. Determination of Kinetic Parameters

Different concentrations (2.5–100 μM) of riboflavin were used to determine the standard enzymatic activity of protein samples. A riboflavin–NBT method was used, and the Michaelis–Menten constant (K_m_) and maximum enzyme velocity (V_max_) of the enzyme was calculated by using the GraphPad prism program.

### 2.15. Statistical Analysis

All experiments were performed at least three times, and the results were expressed as mean ± standard deviations (SD). The statistical significance of the differences was assessed by a one-way ANOVA and followed by least significance difference (LSD) multiple comparison tests. *p* < 0.5 was considered a statistically significant difference.

## 3. Results

### 3.1. Sequence and Homology Analysis of PVCuZnSOD

In this study, a 459 bp ORF, namely, PVCuZnSOD, which encoded a polypeptide of 152 amino acids, was successfully identified. The result of the nucleotide BLAST algorithm showed that PVCuZnSOD was highly similar to CuZnSOD from other sea cucumbers, such as *P. longicauda* (MK580169.1, 93.78% identity), *B. marianensis* (MK580168.1, 71.72% identity), *Paelopatides* sp. (MG959672.1, 72.57% identity), and *A. japonicus* (JX097096.1, 68.60% identity). The results of the SMART program and SignalP program show a predicted domain area from Leu^9^ to Ile^147^ and no signal peptide in PVCuZnSOD. Analysis of the secondary structure by Jalview shows there were nine β-sheets, which have an important influence on the spatial configuration of the protein (Figure 1).

Multiple sequence alignment results showed that the motif ([GA]-[IMFAT]-H-[LIVF]-H-{S}-x-[GP]-[SDG]-x-[STAGDE] (from Gly^42^ to Thr^52^, PS00087), G-[GNHD]-[SGA]-[GR]-x-R-x-[SGAWRV]-C-x(2)-[IV] (from Gly^136^ to Ile^147^, PS00332), and Cu, Zn binding sites (His^44^, His^46^, His^61^, His^69^, His^78^, Asp^81^, and His^118^) were highly conserved (Figure 2). In the phylogenetic tree, PVCuZnSOD was clustered into the invertebrate, which was closely related to Cu.

ZnSOD of other sea cucumber (Figure 3). The three-dimensional model predicted that PVCuZnSOD mostly existed as a homodimer (Figure 4).

**Figure 2 antioxidants-12-01227-f002:**
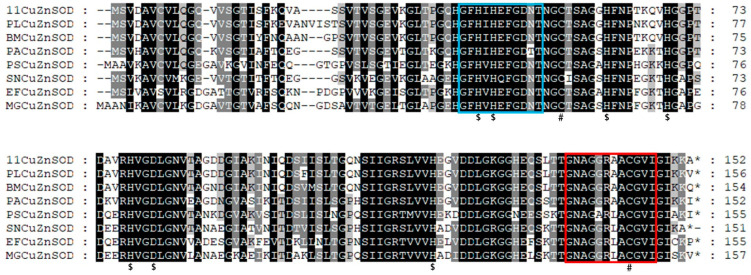
Multiple alignment of PVCuZnSOD with other SODs. The colors of shadow from dark to light represent the conservation of amino acid residues from high to low. Gaps are indicated by dashes to improve the alignment. The GenBank accession numbers are shown in Table 1. ‘*’ represents the terminator of the amino acid sequence. ‘$’ represents metal binding sites, the blue (PS00087) and red (PS00332) box represent motifs. ’#’ represents the two conserved cysteines.

### 3.2. Protein Expression and Purification

The results of the SDS-PAGE (Figure 5A) and Western blot (Figure 5B) showed that the monomer molecular weight of PVCuZnSOD was approximately 15 kDa. IPTG increased the production of recombinant protein. Compared with the non-induced group, the protein production of the IPTG-induced group was higher. In addition, the protein primarily existed in the supernatant after sonication.

### 3.3. Effect of Temperature

The effect of temperature on the enzymatic activity is shown in Figure 6A. PVCuZnSOD maintained a high enzymatic activity in the range of 0 to 60 °C (>75%), and the enzymatic activity reached the highest at 20 °C. After 60 °C, the enzymatic activity dropped sharply, and almost no enzymatic activity was observed at 70 °C.

The stability of PVCuZnSOD was tested under storage (4 °C), application (20 °C and 37 °C), and at a high temperature (50 °C). The result (Figure 6B) showed that PVCuZnSOD activity decreases with the increase in time but maintains >70% maximum enzymatic activity after 6 h at all tested temperatures. After 12 h, PVCuZnSOD incubated at 4 °C, 20 °C, and 37 °C can maintain at least 50% activity, but the activity of PVCuZnSOD incubated at 50 °C decreased to below 50%. After 36 h, PVCuZnSOD incubated at 4 °C and 20 °C only had weak activity, and the groups incubated at 37 °C and 50 °C were completely inactivated.

### 3.4. Effect of pH

The effect of pH on the enzymatic activity is shown in Figure 7. With the increase in pH, the enzymatic activity of PVCuZnSOD initially increases and then decreases, reaching the highest at pH = 11.0. PVCuZnSOD maintains more than 50% activity in the pH range of 4.0–11.0 and more than 65% enzymatic activity in the pH range of 7.0–11.0.

### 3.5. Effect of Metal Ions

The effect of metal ions on the enzymatic activity is shown in Figure 8. At the lower tested concentrations (0.1 and 1 mM) PVCuZnSOD showed strong resistance to almost all metal ions and maintained at least 65% activity. Notably, 0.1 mM and 1 mM Ni^2+^ and Mg^2+^ increased the activity of PVCuZnSOD, whereas 1 mM Cu^2+^ significantly inhibited PVCuZnSOD activity to 24.10%. At a higher tested concentration of 5 mM and 10 mM metal ions, PVCuZnSOD maintains a high residual activity (at least 45%) after being incubated with Ni^2+^, Mg^2+^, Ba^2+^, and Ca^2+^. However, the activity of PVCuZnSOD was strongly inhibited by Mn^2+^, Cu^2+^, Zn^2+^, and Co^2+^. Furthermore, the residual activity completely disappeared under 10 mM Mn^2+^ and Zn^2+^.

### 3.6. Effect of Surfactants

The effect of surfactants on the enzymatic activity is shown in Figure 9A. After treatment with Tween 20 and Triton X-100 of 1%, 5%, and 10% concentrations, the activity of PVCuZnSOD remained above 60%. PVCuZnSOD showed strong tolerance to SDS. After being incubated with 0.1% SDS for 30 min, PVCuZnSOD maintained 60% activity. When SDS concentration reached 0.5%, the activity of PVCuZnSOD remained at 26.85%, and PVCuZnSOD was almost inactivated by 1% SDS.

### 3.7. Effect of Organic Solvents

The activity of PVCuZnSOD was checked with 5% and 10% (*v*/*v*) organic solvents. As shown in Figure 9B, the activity of PVCuZnSOD was evidently not affected by DMSO (5% and 10%). Ethanol, glycerol, and isopropanol (5% and 10%) have a weak inhibitory effect on the activity of PVCuZnSOD, and PVCuZnSOD maintained more than 75% activity. However, after being incubated with methanol and acetone (5% and 10%), the maximum residual activity of PVCuZnSOD was only 54.88%, which indicated that PVCuZnSOD was sensitive to methanol and acetone.

### 3.8. Effect of Denaturants

Urea and GuHCl are strong protein denaturants, and their effects on PVCuZnSOD are shown in Figure 10. Based on the urea curve, the enzymatic activity initially increased, reaching the highest peak (106.79%) at 1 M concentration, and then decreased. After being incubated with 4 M urea, the activity of PVCuZnSOD was decreased to 80.82%. The curve of GuHCl also showed a similar trend to the urea curve; when the concentration of GuHCl increased to 4 M, the activity of PVCuZnSOD was 79.72%.

### 3.9. Effect of Digestive Enzyme

The result of PeptideCutter shows 17 cutting sites of pepsin and 12 cutting sites of trypsin/chymotrypsin in the PVCuZnSOD sequence (Table 2). However, PVCuZnSOD retains high activity after treatment with digestive enzymes. As shown in Table 3, during incubation with pepsin, the activity of PVCuZnSOD gradually decreased to 69% after 3 h treatment. However, when PVCuZnSOD was incubated with trypsin and chymotrypsin, the activity increased to 121.57% at the first hour. Afterward, the enzymatic activity gradually decreased, and at the third hour, the enzymatic activity decreased to 87.96%. By contrast, the activity of bovine SOD decreased rapidly after digestive enzyme treatments: it decreased to 2.18% and 17.85% after 3 h in pepsin and trypsin/chymotrypsin, respectively. Therefore, PVCuZnSOD has stronger anti-digestibility than bovine SOD.

### 3.10. Kinetic Parameters

The kinetic parameters of PVCuZnSOD were determined using different concentrations of riboflavin. The Michaelis-Menten plot was shown in Figure 11. The Km value of riboflavin is 29.24 μM, and the V_max_ value is 1226 units/mg.

## 4. Discussion

Conditions such as high pressure and low temperature in the deep sea can easily cause oxidative stress in organisms [22], leading to cell death and various diseases. SODs are the first line of defense of the biological antioxidant system, which can effectively remove the superoxide anion in the organism and maintain homeostasis in the body. Therefore, SODs from the deep sea may have better performance or special ability.

In this study, PVCuZnSOD was identified from a new species, namely, *P. verruciaudatus*. In PVCuZnSOD, conserved metal binding sites and motifs were identified [23]. In addition, two cysteines (Cys^55^ and Cys^143^) forming disulfide bonds were identified in PVCuZnSOD [24]. These results indicated that PVCuZnSOD was a CuZnSOD homolog. Similarly, the results of the phylogenetic tree also proved that PVCuZnSOD was a CuZnSOD homolog. In general, CuZnSOD can be divided into extracellular CuZnSOD containing an N-terminal signal cleavage peptide and intracellular CuZnSOD without signal peptide, based on localization [25]. Combined with the results of SignalP, PVCuZnSOD is an intracellular Cu-ZnSOD.

Then, the optimum temperature of PVCuZnSOD was detected. The results showed that the enzymatic activity of PVCuZnSOD reached the highest at 20 °C, and the enzymatic activity remained above 85% in the range of 0–60 °C. CuZnSOD from garlic [26] only exhibited a maximum activity of more than 70% in the range of 30–50 °C, and CuZnSOD from Sika deer [27] maintained an enzymatic activity of 85% in the range of 20–50 °C. By contrast, PVCuZnSOD has a wider active temperature range of 0–60 °C. In addition, PVCuZnSOD has a high activity (88.64%) at 40 °C, which is close to human body temperature, indicating that PVCuZnSOD has medicinal potential. PVCuZnSOD also has great thermal stability, which can maintain more than 70% of the maximum activity when incubated at different temperatures (4 °C, 20 °C, 37 °C, and 50 °C) for 6 h. Moreover, PVCuZnSOD maintained more than 50% activity after incubation at 37 °C for 12 h, indicating that PVCuZnSOD can serve as a drug in humans.

In general, acidic conditions will limit the metal–ligand interaction of CuZnSOD and induce protein structure unfolding, thereby inhibiting the activity of CuZnSOD and denaturing the protein. For example, CuZnSOD from *Rimicaris exoculate* [28] is completely inactive at pH 4.0. In addition, the activity is less than 50% at pH 4.0–7.0, and the activity of CuZnSOD from *Liza haematocheila* [29] is less than 20% under acidic conditions. However, PVCuZnSOD maintained a certain activity at pH 1.0–7.0, and the activity was even higher than 50% at pH 4.0–7.0. Therefore, PVCuZnSOD has good acid resistance, indicating that PVCuZnSOD may resist gastric acid conditions, and it has great application potential as an oral drug. Moreover, CuZnSOD has higher activity under alkaline conditions. The enzymatic activity of CuZnSOD from *B. marianensis* sp. nov. reaches the highest at pH 7.4, and its activity decreases rapidly when pH > 9.5 [30]. The optimum pH of CuZnSOD from *P. Longicauda* is 8.0, which drops rapidly at pH > 9.0 [31]. By contrast, PVCuZnSOD activity reached the peak at pH 11 and then dropped rapidly, and it maintained >68.10% activity under pH 7.0–11.0 after a 1 h treatment. The decrease in CuZnSOD activity under alkaline conditions may be due to the deprotonation of basic residues near the active site of the enzyme, reducing the ability to direct superoxide anions to the active site [18]. Moreover, PVCuZnSOD showed a broad pH range, and maintained an enzymatic activity at pH 1.0–13.0; studies have shown that thermostable MnSOD is generally stable over a wide pH range [32]. A similar rule may also exist in CuZnSOD such as CuZnSOD from *Paelopatides* sp. [16], *Curcuma aromatica* [18], and PVCuZnSOD.

Next, the effect of metal ions on the activity of PVCuZnSOD was examined. PVCuZnSOD was not sensitive to most of the tested metal ions, such as Ni^2+^, Mg^2+^, Ba^2+^, and Ca^2+^. The activity of CuZnSOD from *Rimicaris exoculate* [33] was maintained at 57.89% after being incubated with 10 mM Mg^2+^, and CuZnSOD from garlic [26] maintained only 36% activity under 5 mM Mg^2+^. By contrast, after being incubated with 10 mM Mg^2+^ for 30 min, the activity of PVCuZnSOD remained at 83.43%, showing excellent Mg^2+^ tolerance. In general, copper and zinc ions can stabilize CuZnSOD, such as CuZnSOD from human [34] and *Halomonas* sp. ANT108 [35]. The activity of CuZnSOD was improved after incubation with both Cu^2+^ and Zn^2+^. However, in our study, Cu^2+^ and Zn^2+^ inhibit the activity of PVCuZnSOD, which is similar to CuZnSOD from *R. exoculate* [28], *Paelopatides* sp. [16], and *P. longicauda* [31]. These SODs were all obtained from deep sea samples, indicating that copper and zinc ions may have special regulatory mechanisms for CuZnSOD in deep-sea fauna.

The resistance of proteins to surfactants is an important parameter for the kinetic stability of proteins. After being incubated with 10% Tween 20 and 10% TritonX-100, the activity of PVCuZnSOD maintained 62.97% and 76.68%, respectively. Compared with PVCuZnSOD, the SOD from *Geobacillus* sp. EPT3 [18] only maintained 31.5% and 70.3% of the maximum activity after being incubated with 1% Tween 20 and 1% TritonX-100. The SOD from *Rhodotorula mucilaginosa* AN5 [36] was strongly inhibited by 1% TritonX-100, and it maintained less than 30% activity. Therefore, PVCuZnSOD has strong resistance to these two surfactants.

The resistance of proteins to organic solvents is an important factor for evaluating their application potential. PVCuZnSOD maintained at least 75% activity after being treated with ethanol, glycerol, isopropanol, and DMSO (5% and 10% concentrations). These solvents are often used in the production of drugs or daily chemical products, such as glycerin in skin care products [37] and isopropanol in drug production [38]. Therefore, PVCuZnSOD has a broad application potential in drugs and daily chemical products. In addition, some organic solvents such as methanol and acetone strongly inhibited the enzymatic activity of PVCuZnSOD, which may depend on the electronic configuration of the residues in the active site of PVCuZnSOD and the hydrogen bonds between the different residues and the metal ion-coordinated hydrogen bonding network [39].

In this study, the activity of PVCuZnSOD maintained more than 80% activity after being incubated with 4 M urea for 30 min, showing strong resistance to urea. Based on previous studies, CuZnSOD seems to have strong resistance to urea. Franco et al. pointed out that urea causes the instability of the CuZnSOD dimer structure, which has no significant impact on other properties. PVCuZnSOD maintained 76.01% activity at 4 M concentration for 30 min, showing strong resistance to GuHCl.

Digestive enzymes break the specific amino acid sites in the protein and turn the complete protein into inactive polypeptide chains, thereby inactivating the protein. Some CuZnSOD showed strong resistance to digestive enzymes, such as CuZnSOD from lemon [40] and *Epinephelus malabaricus* [41]. In this study, although the PeptideCutter detected several potential cutting sites in PVCuZnSOD, PVCuZnSOD still retained high activity in the experiment. This result may be due to the stable conformation of the active center of PVCuZnSOD, or the cleavage of the cutting site does not affect the structure of the active center. In addition, the combined treatment of trypsin and chymotrypsin temporarily increases the activity of PVCuZnSOD. Trypsin and chymotrypsin destroy the outer structure of PVCuZnSOD protein and expose its active center, thereby improving the scavenging efficiency of active oxygen in a short time. The hydrolysis of SODs by digestive enzymes is one of the main concerns affecting the application of SODs. The excellent resistance of PVCuZnSOD to digestive enzymes showed its potential as an oral drug or dietary additive.

Our research has obtained new CuZnSOD with high stability and great application potential, which can be applied to medical, cosmetics, food, and other industries.

## 5. Conclusions

In this study, one CuZnSOD gene from *P. verruciaudatus* was cloned and characterized. This CuZnSOD has a wide range of pH activity, cold resistance, and strong resistance to most metal ions, surfactants, organic solvents, denaturants, and digestive enzymes. These properties indicate that PVCuZnSOD has a broad application potential in the fields of bioengineering, cosmetics, and medicine.

## Figures and Tables

**Figure 1 antioxidants-12-01227-f001:**
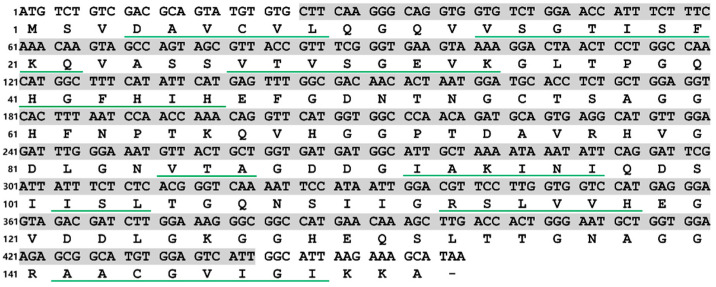
Nucleotide and deduced amino acid sequences of PVCuZnSOD. The green underlines represent β sheets. The shaded part from Leu^9^ to Ile^147^ represents the predicted domain area.

**Figure 3 antioxidants-12-01227-f003:**
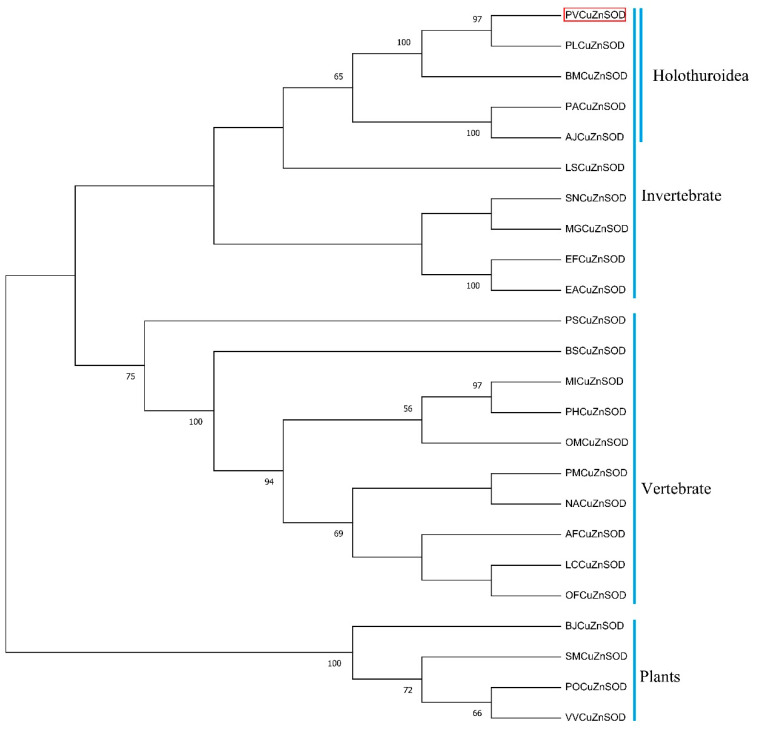
Phylogenetic trees for PVCuZnSOD constructed by the neighbor-joining method based on the sequences from different animals. The PVCuZnSOD was highlight by the red box. The numbers before the branches (only display >50%) in the figure represent confidence. The Genbank accession numbers are shown in Table 1.

**Figure 4 antioxidants-12-01227-f004:**
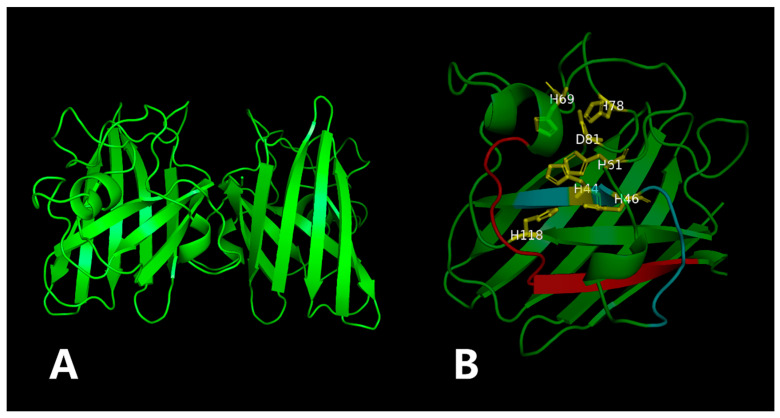
The three-dimensional model of the PVCuZnSOD. The yellow part is the metal binding site, and the red (PS00332) and blue (PS00087) part is the location of the motif. (**A**) 3D structure of PVCuZnSOD using the Swiss model. (**B**) Close-up of the active sites.

**Figure 5 antioxidants-12-01227-f005:**
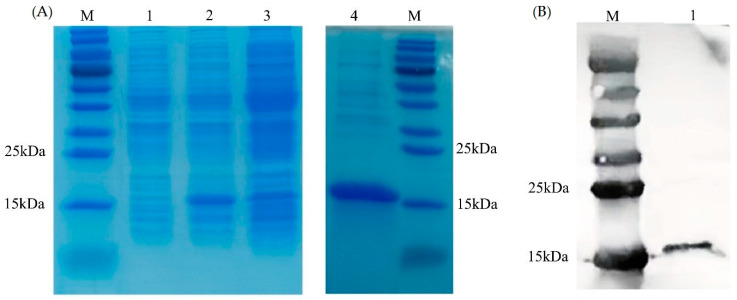
Expression and purification of PVCnZnSOD. (**A**) SDS-PAGE analysis of PVCuZnSOD. Lane M: marker. Lane 1: total protein of *E. coli* pCold II/BL21 containing recombinant plasmid before IPTG induction. Lane 2: total protein of *E. coli* pCold II/BL21 containing recombinant plasmid after IPTG induction. Lane 3: supernatant after ultrasonic treatment. Lane 4: purified recombinant protein. (**B**) Western blot analysis of PVCuZnSOD. Lane M: marker. Lane 1: purified recombinant protein.

**Figure 6 antioxidants-12-01227-f006:**
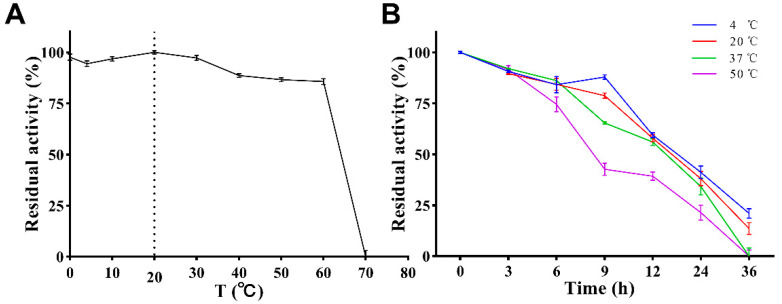
Effect of temperature on activity of PVCuZnSOD. (**A**) Effect of temperature on activity of PVCuZnSOD. The dotted line represents the temperature at which the enzyme activity is highest. (**B**) Effect of temperature on stability of PVCuZnSOD.

**Figure 7 antioxidants-12-01227-f007:**
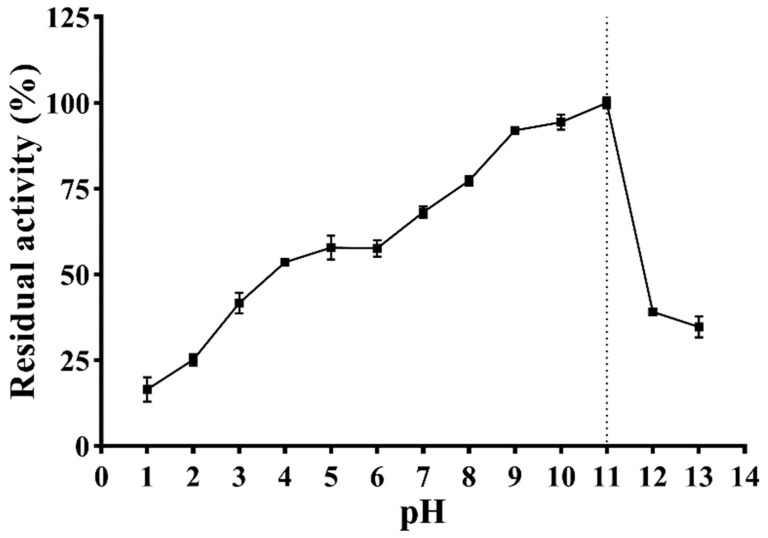
Effect of pH on activity of PVCuZnSOD. The dotted line represents the pH corresponding to the highest enzymatic activity.

**Figure 8 antioxidants-12-01227-f008:**
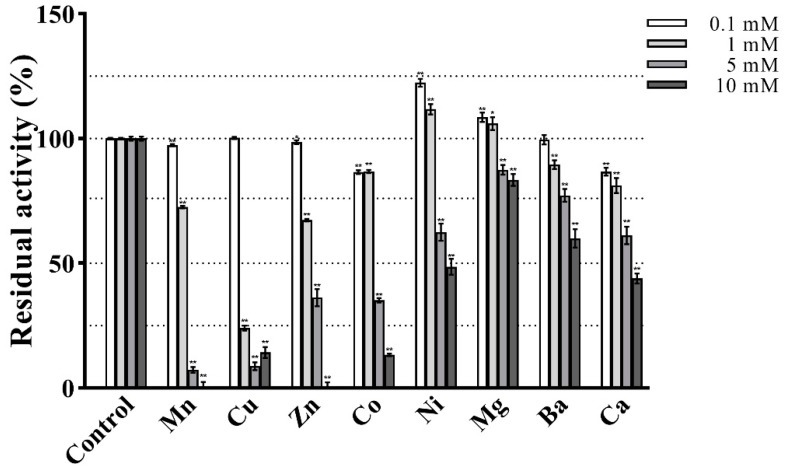
Effect of various treatments on the activity of PVCuZnSOD. * *p* < 0.05, ** *p* < 0.01 compared with the control group.

**Figure 9 antioxidants-12-01227-f009:**
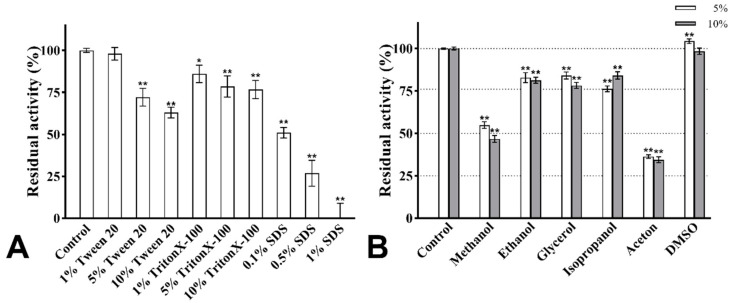
The effect of surfactants (**A**) and organic solvents (**B**). * *p* < 0.05, ** *p* < 0.01 compared with the control group.

**Figure 10 antioxidants-12-01227-f010:**
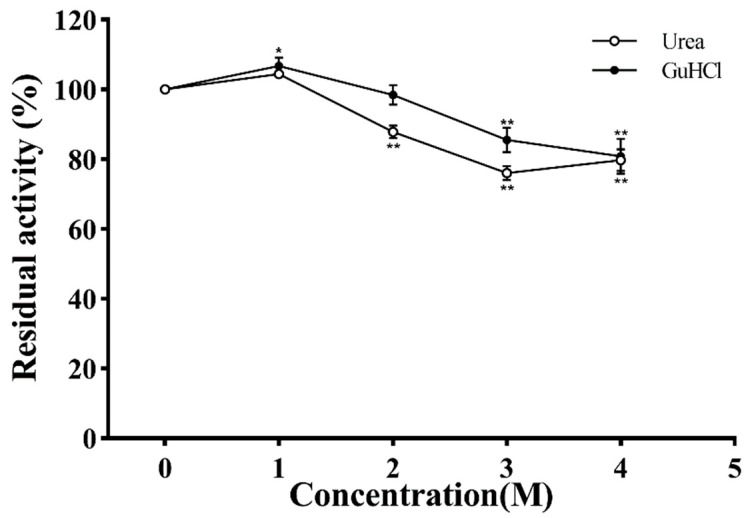
The effect of urea and GuHCl. * *p* < 0.05, ** *p* < 0.01 compared with the control group.

**Figure 11 antioxidants-12-01227-f011:**
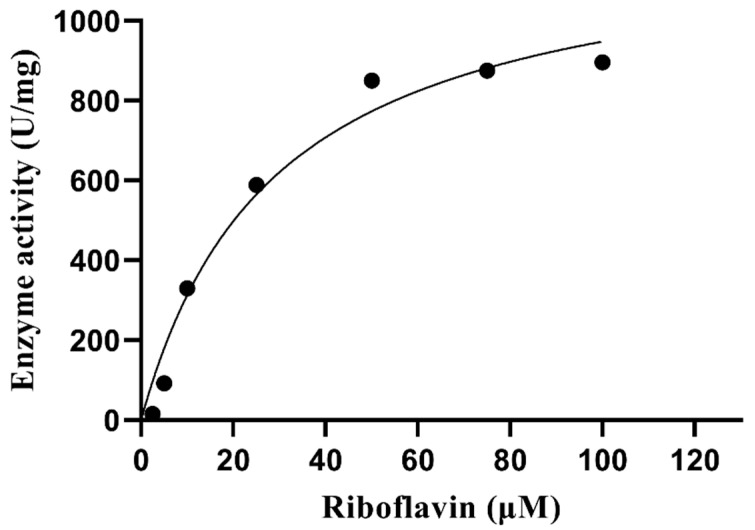
Michaelis–Menten plot of PVCuZnSOD.

**Table 1 antioxidants-12-01227-t001:** Sequences used for multiple alignments and phylogenetic analysis.

Genbank	Species	Abbreviation
OQ687337	*P. verruciaudatus*	PVCuZnSOD
MK580169.1	*Psychropotes longicauda*	PLCuZnSOD
MK580168.1	*Benthodytes marianensis*	BMCuZnSOD
MG989672.1	*Paelopatides* sp.	PACuZnSOD
JX097096.1	*Apostichopus japonicus*	AJCuZnSOD
KT594770.1	*Sterechinus neumayeri*	SNCuZnSOD
MF324875.1	*Eisenia fetida*	EFCuZnSOD
KR106132.1	*Eisenia andrei*	EACuZnSOD
FM177867.1	*Mytilus galloprovincialis*	MGCuZnSOD
AY332385.2	*Lymnaea stagnalis*	LSCuZnSOD
JX470524.1	*Pelodiscus sinensis*	PSCuZnSOD
BT082811.1	*Anoplopoma fimbria*	AFCuZnSOD
JN032591.1	*Amphiprion clarkii*	ACCuZnSOD
AF329278.1	*Pagrus major*	PMCuZnSOD
MH716024.1	*Mugil incilis*	MICuZnSOD
AY491056.1	*Oreochromis mossambicus*	OMCuZnSOD
HQ318825.1	*Lates calcarifer*	LCCuZnSOD
AY613390.1	*Oplegnathus fasciatus*	OFCuZnSOD
MG208871.1	*Nibea albiflora*	NACuZnSOD
MK860772.1	*Planiliza haematocheilus*	PHCuZnSOD
MT123896.1	*Bostrychus sinensis*	BSCuZnSOD
X95726.1	*Brassica juncea*	BJCuZnSOD
DQ481231.1	*Populus suaveolens*	POCuZnSOD
KU240390.1	*Solanum melongena*	SMCuZnSOD
JQ692111.2	*Vitis vinifera*	VVCuZnSOD

**Table 2 antioxidants-12-01227-t002:** The cutting sites of digestive enzyme.

Name of Enzyme	No. of Cleavages	Positions of Cleavage Sites
Pepsin (pH > 2)	17	8 9 19 20 35 42 47 61 81 82 103 104 114 123 124 132 133
Chymotrypsin	2	20 48
Trypsin	10	21 34 66 77 94 113 126 141 150 151

**Table 3 antioxidants-12-01227-t003:** The effect of digestive enzymes.

		Residual Activity (%)	
	Time (h)	Bovine SOD	PVCuZnSOD
Pepsin	0	100.00 ± 2.05	100.00 ± 0.75
	1	88.21 ± 3.86	93.21 ± 0.70
	2	42.29 ± 4.86	81.28 ± 1.43
	3	2.18 ± 8.29	69.59 ± 1.16
Trypsin/chymotrypsin	0	100.00 ± 4.94	100.00 ± 2.27
	1	75.79 ± 1.10	121.57 ± 2.30
	2	43.92 ± 4.11	105.73 ± 3.29
	3	17.85 ± 6.20	87.96 ± 7.50

## Data Availability

All experimental supporting data and procedures are available within this article.

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
