# Peer review of "Characterization of a Novel Superoxide Dismutase from a Deep-sea Sea Cucumber (Psychoropotes verruciaudatus)"

_antioxidants, 2023, doi:10.3390/antiox12061227_

Round 1
Reviewer 1 Report
The manuscript by Li et al. describes the identification of a superoxide-dismutase-encoding gene from a sea cucumber from a deep-sea environment, its recombinant expression and the characterization of the gene product. The experimental work is straight-forward and by and large well described (with some shortcomings, details follow), and the manuscript is mostly well written. There are, however, a number of aspects that require improvement to make the manuscript publishable.
In detail: the Introduction shows some linguistic flaws, a revision by a native speaker of English is recommended. The description of the effect of ROS/oxidative stress on the (human) body as well as the applications in medical or cosmetic products is superficial – how does “oxidative stress” affect the body or health, what are the consequences, and how can preparations or formulations based on SOD alleviate those? What is the effect of SOD in medical drugs or cosmetics or food products, are there examples of such products, how is SOD formulated to be included in such products? There is no need for a detailed review, but a bit more than the few cursory sentences presented here are required for a better understanding of the applicational potential – particularly with respect to a new enzyme from a rather exotic source. For example, are there particular shortcomings to the enzymes that are currently used or considered for such applications that need improvements, or even entirely new enzymes? Two examples from recent works on SOD from deep-sea organisms are a bit thin to make that case, and the Results and Discussion sections later do not refer to particular advantages of the new enzyme. In the Discussion, the suitability of the enzyme in drugs for humans is mentioned as well, but it remains unclear what kind of drug that should be, and what diseases or syndromes should be treated with it. In this context, the meaning of the experiments using digestive enzymes remains obscure as well – would an SOD that is included in a medicinal drug be exposed to digestive enzymes? How would such a drug be formulated, where would be the designated location of the therapeutic activity?
In the Introduction, the source organisms from other enzymes are only mentioned by their scientific names, it would be helpful to at least mention whether these life forms are reptiles, or mammals, or invertebrates.
Line 382f (Discussion): “copper and zinc ions may have special regulatory mechanisms for CuZnSOD in deep-sea fauna” – that is speculative and not substantiated by any other supportive data or a hypothesis.
I think the small size of the enzyme deserves a mention, and a comparison with other related enzymes.
Minor points:
Line 87, the spelling of BamH1 and PstI is wrong.
Line 100, was sonication really performed continuously for one hour?
Line 102, where does the His-tag come from, that was used for purification? Does the pCOLDII vector include a His-tag in frame when these two restriction enzymes are used for insertion? A clearer explanation of the used system is in order here.
Line 115, what antibody was used? Is it directed against the His-tag?
Line 202, “..shows a night β-sheet, which has an important influence...“ – what is a night β-sheet? Please correct.
Line 231. “the protein primarily existed in the supernatant” is misleading – this would mean it is a secretory protein, which was refuted earlier. “Supernatant” obviously refers to the supernatant after sonication – please correct.
Line 326, “oxidative stress easily occurs in organisms” (in the deep sea) – why?
Line 333, cysteins forming disulfide bonds – disulfide bonds are formed in a reductive environment, not in the oxidative environment of the cytoplasm, occur primarily in secretory proteins and are found in cytoplasmic proteins only in exceptions. What is the evidence that these disulfide bonds are actually formed (MS analysis of digested peptides would help)? Is this only inferred from the structural model? This needs to be explained.
There are some weaknesses in the Introduction, which could use a revision by a native English speaker.
Author Response
8 May. 2023
Dear editor,
Thanks for the reviews of our manuscript. We have revised it carefully based on helpful suggestions of reviewers. Responses to each suggestion are provided below. Please let me know if there are any further questions or concerns.
Thank you for your consideration.
Sincerely,
Yanan Li,
Response to Reviewer #1
The manuscript by Li et al. describes the identification of a superoxide-dismutase-encoding gene from a sea cucumber from a deep-sea environment, its recombinant expression and the characterization of the gene product. The experimental work is straight-forward and by and large well described (with some shortcomings, details follow), and the manuscript is mostly well written. There are, however, a number of aspects that require improvement to make the manuscript publishable.
In detail: the Introduction shows some linguistic flaws, a revision by a native speaker of English is recommended. The description of the effect of ROS/oxidative stress on the (human) body as well as the applications in medical or cosmetic products is superficial – how does “oxidative stress” affect the body or health, what are the consequences, and how can preparations or formulations based on SOD alleviate those? What is the effect of SOD in medical drugs or cosmetics or food products, are there examples of such products, how is SOD formulated to be included in such products? There is no need for a detailed review, but a bit more than the few cursory sentences presented here are required for a better understanding of the applicational potential – particularly with respect to a new enzyme from a rather exotic source. For example, are there particular shortcomings to the enzymes that are currently used or considered for such applications that need improvements, or even entirely new enzymes? Two examples from recent works on SOD from deep-sea organisms are a bit thin to make that case, and the Results and Discussion sections later do not refer to particular advantages of the new enzyme. In the Discussion, the suitability of the enzyme in drugs for humans is mentioned as well, but it remains unclear what kind of drug that should be, and what diseases or syndromes should be treated with it. In this context, the meaning of the experiments using digestive enzymes remains obscure as well – would an SOD that is included in a medicinal drug be exposed to digestive enzymes? How would such a drug be formulated, where would be the designated location of the therapeutic activity?
Thanks for your suggestion. The present manuscript has been revised by a native speaker of English. We have improved the introduction on how oxidative stress affects the body or health and the role of SOD in medical drugs, cosmetics, or food. In addition, we describe more about the shortcomings of the current application of SODs. Experiments with digestive enzymes were conducted to explore the stability of PVCuZnSOD and to evaluate its potential for oral administration. The preparation of PVCuZnSOD as a drug and the designated location of its therapeutic activities are also a focus of our future research.
Line 382f (Discussion): “copper and zinc ions may have special regulatory mechanisms for CuZnSOD in deep-sea fauna” – that is speculative and not substantiated by any other supportive data or a hypothesis.
Thanks for your suggestion. Copper and zinc ions will increase the activity of SOD in many studies, while the activity of SOD from deep-sea organisms is often inhibited by copper and zinc ions, which has appeared in multiple studies. So, based on these previous studies, we gave such speculation.
Minor points:
Line 87, the spelling of BamH1 and PstI is wrong.
The spelling of BamH1 and PstI has been corrected.
Line 100, was sonication really performed continuously for one hour?
Sonication operates continuously for 5 seconds with an interval of 3 seconds. We have also made corresponding modifications in the chapter 2.4
Line 102, where does the His-tag come from, that was used for purification? Does the pCOLDII vector include a His-tag in frame when these two restriction enzymes are used for insertion? A clearer explanation of the used system is in order here.
His-tag is included in the pCOLDII vector, the recombinant plasmid also contains His-tag and we have supplemented with explanations in part 2.3.
Line 115, what antibody was used? Is it directed against the His-tag?
The primary antibody was Anti-His Tag Mouse Monoclonal Antibody and the secondary antibody was HRP Goat Anti-Mouse lgG. We have added antibody information in part 2.5.
Line 202, “..shows a night β-sheet, which has an important influence...“ – what is a night β-sheet? Please correct.
The description in the original text is incorrect, “night” has been modified to “nine”.
Line 231. “the protein primarily existed in the supernatant” is misleading – this would mean it is a secretory protein, which was refuted earlier. “Supernatant” obviously refers to the supernatant after sonication – please correct.
“Supernatant” has been modified to “supernatant after sonication”.
Line 326, “oxidative stress easily occurs in organisms” (in the deep sea) – why?
The reason why the deep sea is prone to oxidative stress is due to its extreme environment such as low temperature and high pressure[1]. These adverse conditions will cause the mass production of ROS in the body, leading to oxidative stress.
Line 333, cysteins forming disulfide bonds – disulfide bonds are formed in a reductive environment, not in the oxidative environment of the cytoplasm, occur primarily in secretory proteins and are found in cytoplasmic proteins only in exceptions. What is the evidence that these disulfide bonds are actually formed (MS analysis of digested peptides would help)? Is this only inferred from the structural model? This needs to be explained.
Thanks for your suggestion. The existence of disulfide bonds is a result obtained from previous studies, and there is indeed a unique disulfide bond in CuZnSOD. Some researchers believe that this is also a feature that can be used to identify CuZnSOD[2].
References:
- Xiao, X.; Zhang, Y., Life in extreme environments: Approaches to study life-environment co-evolutionary strategies. SCIENCE CHINA Earth Sciences 2014, (5), 9.
- Sea, K.; Sohn, S. H.; Durazo, A.; Sheng, Y.; Shaw, B. F.; Cao, X.; Taylor, A. B.; Whitson, L. J.; Holloway, S. P.; Hart, P. J., Insights into the role of the unusual disulfide bond in copper-zinc superoxide dismutase. Journal of Biological Chemistry 2015, 290, (4), 2405-2418.

Reviewer 2 Report
In the present work, the authors claim to have cloned, heterologously expressed, and characterized a novel copper-zinc superoxide dismutase (CuZnSOD) from a new species of sea cucumber (Psychropotes verrucicaudatus) with "great application potential in medicine, food, and other products". While the recombinant enzyme was generally well characterized biochemically, some improvements were suggested.
In the Introduction, the authors neither explain nor clarify why they consider "CuZnSOD" to have "great application potential in medicine, food, and other products". The Introduction should be improved to reference the need for new and better SOD enzymes in medicine and food. In the Discussion, the authors should specify how "CuZnSOD" can fill the need for SOD in these areas.
In Materials and Methods, in lines 84 and 85, the authors claim to have amplified PVCuZnSOD by polymerase chain reaction (PCR) with the primers F: CGGGATCCATGTCTGTCGACGCAGTA and R: GCTGCAGTTATGCTTTCTTAATGCC. However, it was not clear what previous information the authors used to design these specific primers. If the gene coding for PVCuZnSOD was obtained through a previous analysis, the authors should reference the accession number of the sequence or the accession number of the bioproject that includes the coding sequence (e.g. RNAseq). If the coding sequence is completely new, the authors should explain how they designed the primers.
In lines 108 and 109, the authors claim to have evaluated the purification and expression effects of recombinant PVCuZnSOD using SDS-PAGE. The authors should reference the percentage of gel used.
In lines 115 and 116, the authors describe using a primary antibody (Abcam, Cambridge, UK) overnight and a secondary antibody (Abcam, Cambridge, UK) for 2 h. However, the authors omit what kind of primary and secondary antibodies were used.
In the Results section, if PVCuZnSOD is a new sequence, the authors should submit the sequence of PVCuZnSOD to a public database, and they must reference the accession number of the submitted sequence. In "3.2. Protein expression and purification," the authors should reference expression yields.
In lines 209 and 210, the authors mention that "The three-dimensional model predicted that PVCuZnSOD mostly existed as a homodimer (Fig. 4)." However, the authors fail to explain the active site of this enzyme, despite the fact that the amino acids of the active site are represented in Figure 4. The SWISS-MODEL used by the authors uses a comparative approach and a database of annotated models. The authors should provide information about the SOD proteins used to generate PVCuZnSOD's predicted structure.
In Figure 2, the multiple alignment of PVCuZnSOD with other SODs, the authors should identify the two "conserved" cysteines (Cys55 and Cys143) forming disulfide bonds.
In Figure 3, the phylogenetic trees for PVCuZnSOD constructed by the neighbor-joining method, the authors should provide the level of statistical significance for each branching point in the tree based on the proportion of bootstrap support.
In Figure 6, Graph B, the lines should have different symbols or markers to facilitate discrimination.
Author Response
8 May. 2023
Dear editor,
Thanks for the reviews of our manuscript. We have revised it carefully based on helpful suggestions of reviewers. Responses to each suggestion are provided below. Please let me know if there are any further questions or concerns.
Thank you for your consideration.
Sincerely,
Yanan Li,
Response to Reviewer #2
In the Introduction, the authors neither explain nor clarify why they consider "CuZnSOD" to have "great application potential in medicine, food, and other products". The Introduction should be improved to reference the need for new and better SOD enzymes in medicine and food. In the Discussion, the authors should specify how "CuZnSOD" can fill the need for SOD in these areas.
Thanks for your suggestion. We have improved in the introduction on how oxidative stress affects the body or health, as well as the role of SOD in medical drugs, cosmetics, or food.
In Materials and Methods, in lines 84 and 85, the authors claim to have amplified PVCuZnSOD by polymerase chain reaction (PCR) with the primers F: CGGGATCCATGTCTGTCGACGCAGTA and R: GCTGCAGTTATGCTTTCTTAATGCC. However, it was not clear what previous information the authors used to design these specific primers. If the gene coding for PVCuZnSOD was obtained through a previous analysis, the authors should reference the accession number of the sequence or the accession number of the bioproject that includes the coding sequence (e.g. RNAseq). If the coding sequence is completely new, the authors should explain how they designed the primers.
Our primers were designed based on the nucleotide sequence of the PVCuZnSOD (Accession number: OQ687337).
In lines 108 and 109, the authors claim to have evaluated the purification and expression effects of recombinant PVCuZnSOD using SDS-PAGE. The authors should reference the percentage of gel used.
The purification and expression effects of recombinant PVCuZnSOD by 12% SDS-PAGE gel. We have added it in our manuscript of chapter 2.4.
In lines 115 and 116, the authors describe using a primary antibody (Abcam, Cambridge, UK) overnight and a secondary antibody (Abcam, Cambridge, UK) for 2 h. However, the authors omit what kind of primary and secondary antibodies were used.
The primary antibody was Anti-His Tag Mouse Monoclonal Antibody and the secondary antibody was HRP Goat Anti-Mouse lgG. We have revised it in chapter 2.5.
In the Results section, if PVCuZnSOD is a new sequence, the authors should submit the sequence of PVCuZnSOD to a public database, and they must reference the accession number of the submitted sequence.
Thanks for your suggestion. The sequence of PVCuZnSOD was submitted to NCBI (Accession number: OQ687337).
In "3.2. Protein expression and purification," the authors should reference expression yields.
Thanks for your suggestion. Our present work did not record the expression yields data and we will pay attention to this issue in future work.
In lines 209 and 210, the authors mention that "The three-dimensional model predicted that PVCuZnSOD mostly existed as a homodimer (Fig. 4)." However, the authors fail to explain the active site of this enzyme, despite the fact that the amino acids of the active site are represented in Figure 4. The SWISS-MODEL used by the authors uses a comparative approach and a database of annotated models. The authors should provide information about the SOD proteins used to generate PVCuZnSOD's predicted structure.
The structure prediction of PVCuZnSOD was based on the most appropriate model ((PDB ID:7wx0.1.A Superoxide dismutase [Cu-Zn], E40K variant of Cu/Zn-superoxide dismutase from dog). The information has been supplemented in chapter 2.2.
In Figure 2, the multiple alignment of PVCuZnSOD with other SODs, the authors should identify the two "conserved" cysteines (Cys55 and Cys143) forming disulfide bonds.
Two conserved cysteines (Cys55 and Cys143) forming disulfide bonds has been marked in Fig 2.
In Figure 3, the phylogenetic trees for PVCuZnSOD constructed by the neighbor-joining method, the authors should provide the level of statistical significance for each branching point in the tree based on the proportion of bootstrap support.
The confidence level of the branch of the phylogenetic tree has been supplemented in Fig.3.
In Figure 6, Graph B, the lines should have different symbols or markers to facilitate discrimination.
Thanks for your suggestion. The image will be published in color, and the lines will be distinguished by different colors.
Reviewer 3 Report
This paper purified CuZnSOD from new species of sea cucumber in the deep sea, and investigated its enzymatic properties. Oxidative stress, involved in many diseases, is defined as an impaired between reactive oxygen species (ROS) production and antioxidant defense. Especially, antioxidant enzyme, including superoxide dismutase, catalase and peroxidase, play a key role in diminishing oxidative stress. The removal of ROS by exogenous SOD derived from may be an effective preventive strategy against various diseases caused by ROS.
However, I think that the application of exogenous SOD in medicine, cosmetics and food supplement has a lot problems as following;
1) A poor bioavailability of exogenous SOD
2) How to delivery SOD to the target
3) How to avoid the immune system in body and skin
4) In food supplement, the native structure and activity are not maintained in digestive system and the peptides resulted from gastro-intestinal digestion may be absorbed?
Nevertheless, this paper suggests the SOD derived from the new sea cucumber, and investigates the valuable results for its’ properties
Please, check the following indications
Line 97: Centrifugal force is different according centrifuge and rotor type. Please, change rpm to g force.
Line 105: What does pH of elution buffer?
Line 122: Sentence structure is strange; The PVCuZnSOD is assayed by a hydroxylamine method using the total ….. ?
Line 148: Sentence structure is strange; ~ was assayed by a hydroxylamine method?
Line 228-232: Whydo you suggest the purity of protein? You suggest that PVCuZnSOD is composed of dimer. Why do you that using non-denaturing electrophoresis?
Fig 6B. All figures will be published as color? If it is not, the legend of Fig.6B was represented as the different shape.
Line 273: Why are the PVCuZnSOD activities inhibited in the high concentration of Mn, Cu and Zn ion ? They affect the binding capacity between substrate and enzyme?
Fig.9. Do you have a reason to use Michaelis-Meten plot instead of Lineweaver-Burk plot for measuring of Km and Vmax?
Line 329, Line 419-420: The development of strategy to delivery SOD to target is needed for the applications. Further investigations need to be carried out to test the hypothesis that SODs supplementation acts by inducing an endogenous antioxidant defense.
Author Response
8 May. 2023
Dear editor,
Thanks for the reviews of our manuscript. We have revised it carefully based on helpful suggestions of reviewers. Responses to each suggestion are provided below. Please let me know if there are any further questions or concerns.
Thank you for your consideration.
Sincerely,
Yanan Li,
Response to Reviewer #3
Line 97: Centrifugal force is different according centrifuge and rotor type. Please, change rpm to g force.
The rpm has been changed to g force as follow: 8000 rpm→ 4200 × g, 6000 rpm→ 4200 × g, 12000 rpm→ 15000 × g.
Line 105: What does pH of elution buffer?
The pH of elution buffer is 8.0 and we have added it.
Line 122: Sentence structure is strange; The PVCuZnSOD is assayed by a hydroxylamine method using the total ….. ?
The structure of the sentence has been changed in part 2.6, and the results are as follows: The activity of PVCuZnSOD was assay by the total superoxide dismutase assay kit (Nanjing Jiancheng Bioengineering Institute) following the manufacturer’s instructions.
Line 148: Sentence structure is strange; ~ was assayed by a hydroxylamine method?
We have revised it.
Line 228-232: Why do you suggest the purity of protein? You suggest that PVCuZnSOD is composed of dimer. Why do you that using non-denaturing electrophoresis?
We conducted a rough evaluation of the purity of the protein using SDS-PAGE. Both denaturing and non-denaturing electrophoresis are usually used to identify protein expression condition. According to bioinformatics results, the molecular weight of PVCuZnSOD monomer is about 15kDa, and the result on SDS-page also prove it.
Fig 6B. All figures will be published as color? If it is not, the legend of Fig.6B was represented as the different shape.
All figures will be published as color.
Line 273: Why are the PVCuZnSOD activities inhibited in the high concentration of Mn, Cu and Zn ion ? They affect the binding capacity between substrate and enzyme?
Thanks for your suggestion. This is only our current preliminary results, and the specific mechanism needs to be determined.
Fig.9. Do you have a reason to use Michaelis-Meten plot instead of Lineweaver-Burk plot for measuring of Km and Vmax?
Both Michaelis-Meten plot and Lineweaver-Burk plot are common methods for measuring Km and Vmax, and we choose Michaelis-Meten plot because this method is frequently used in other studies, references are as follows:
- Shahi, Z. K. M.; Takalloo, Z.; Mohamadzadeh, J.; Sajedi, R. H.; Haghbeen, K.; Aminzadeh, S., Thermophilic iron containing type superoxide dismutase from Cohnella sp. A01. International Journal of Biological Macromolecules 2021, 187, 373-385.
- Kumar, A.; Kaachra, A.; Bhardwaj, S.; Kumar, S., Copper, zinc superoxide dismutase of Curcuma aromatica is a kinetically stable protein. Process Biochemistry 2014, 49, (8), 1288-1296.
- Wang, Q.; Nie, P.; Hou, Y.; Wang, Y., Purification, biochemical characterization and DNA protection against oxidative damage of a novel recombinant superoxide dismutase from psychrophilic bacterium Halomonas sp. ANT108. Protein Expression and Purification 2020, 173, 105661.
Line 329, Line 419-420: The development of strategy to delivery SOD to target is needed for the applications. Further investigations need to be carried out to test the hypothesis that SODs supplementation acts by inducing an endogenous antioxidant defense.
Thanks for your suggestions. In this study, we focus on the dynamic properties of PVCuZnSOD. The application of PVCuZnSOD is a new research area and we will focus on it in future work.